# Plasma Endocannabinoid Levels in Patients with Borderline Personality Disorder and Healthy Controls

**DOI:** 10.3390/ijms242417452

**Published:** 2023-12-14

**Authors:** Jennifer Spohrs, Valentin Kühnle, David Mikusky, Niklas Sanhüter, Ana Macchia, Sandra Nickel, Birgit Abler

**Affiliations:** 1Department for Child and Adolescent Psychiatry and Psychotherapy, Ulm University Medical Centre, 89075 Ulm, Germany; jennifer.spohrs@uni-ulm.de; 2Department of Psychiatry, Psychotherapy and Psychotraumatology, Military Medical Centre, 89081 Ulm, Germany; 3Department of Psychiatry and Psychotherapy III, Ulm University Medical Centre, 89075 Ulm, Germany; valentin.kuehnle@uni-ulm.de (V.K.); david.mikusky@uni-ulm.de (D.M.); niklas.sanhueter@uni-ulm.de (N.S.); ana.macchia@uni-ulm.de (A.M.); sandra.nickel@uni-ulm.de (S.N.)

**Keywords:** borderline personality disorder, endocannabinoid system, anandamide, *FAAH* rs324420, depression

## Abstract

Borderline personality disorder (BPD) is a highly prevalent psychiatric disorder and presents a complex therapeutic challenge due to limited treatment modalities. Recent focus has converged on the endocannabinoid system (ECS) as a prospective modulator of psychopathological processes in BPD. To address this hypothesis, we analysed plasma endocannabinoid concentrations, specifically anandamide (AEA) and 2-arachidonoylglycerol (2-AG), in a cohort of 49 female BPD patients and 32 matched healthy controls (HC). Additionally, we examined the effect of the *FAAH* polymorphism rs324420 and correlates with psychopathology. The results indicate heightened AEA levels and, by trend, augmented 2-AG levels within the patient group, as compared to the HC group. Significant between group differences in AEA levels were evident in the CC genotype (*FAAH_*rs324420) but not in A-allele carriers while the commonly observed difference in AEA levels between A-allele carriers as compared to the CC genotype was not evident in patients. An effect of genotype was found with higher ratings of depression (Beck’s depression inventory, BDI-II) in the CC genotype compared to A-allele carriers (*FAAH*_rs32442), particularly in the patients. Significant alterations in AEA (and by trend in 2-AG) in patients with BPD may relate to compensatory ECS activity. The finding that the effect is most pronounced in CC homozygotes, might point towards a countermeasure to balance physiologically lower baseline AEA levels. The findings warrant further research to develop potentially beneficial psychopharmacological therapies.

## 1. Introduction

Borderline Personality Disorder (BPD) is a highly prevalent psychiatric disorder, which affects approximately 3% of the population worldwide [1]. It is characterised by a severe psychopathology, leading to persistent difficulties in emotion regulation, impulsivity, suicidal ideations, chronic feelings of emptiness, disturbances in self-perception and non-suicidal self-injury [2]. In terms of the cost of treatment and decrease in productivity, it is among the psychiatric disorders that pose the greatest burden to society [3].

The aetiology of BPD is based on a multifactorial model subsuming biopsychosocial factors. The psychological factors comprise low stress tolerance, low self-esteem, low self-confidence, and a high level of self-criticism [4]. Among the social and environmental factors are a lack of social support, mobbing, a negative influence of social media, and aversive experiences during early childhood [5,6,7]. Furthermore, patients frequently report interpersonal traumatic experiences during childhood and youth, such as sexual abuse or domestic violence [8,9,10]. As biological factors, an altered cortisol response in situations of social stress, a higher pain threshold and tolerance, changes in emotion regulation, and the adjunctive front-limbic circuitry have been identified [11,12,13,14]. Here, neuroimaging studies have found a dysregulation in the amygdala and the anterior cingulate cortex (ACC) in tasks involving emotion processing [15]. Even though a lot of research has been conducted in the past decades, the neurochemical mechanisms involved in the pathology are still only partly understood. Several studies point towards a dysfunction of different neuroendocrinological systems, including the serotonergic and dopaminergic system [16,17,18], but translational research remains scarce. However, in animal and human studies, the endocannabinoid system (ECS) has been shown to be involved in many related processes, such as emotion regulation, consolidation of aversive memories, motivation, pain perception and stress homeostasis [19,20,21]. As such, elevated endocannabinoid concentrations have been found to play a protective role in the development of anxiety [22] and to facilitate extinction learning and extinction retention [23,24,25]. Given that BPD is often linked to traumatic experiences during childhood and youth, it can be assumed that endocannabinoid signalling may be linked to the development, persistence, and therapy of the disorder [26,27,28,29,30,31,32]. This link seems also apparent since many patients consume cannabis as a form of self-medication [33,34].

The ECS consists of two cannabinoid receptors (CB1 and CB2), as well as their ligands. Besides the name giving 9-delta-tetrahydrocannabinol (THC) as an exogenous psychoactive substance, the endocannabinoids anandamide (AEA) and 2-arachidonoylgylcerol (2-AG) play an important role in the above listed processes. Depending on synaptic activation, AEA and 2-AG are synthesised on demand in the postsynaptic neuron following membrane depolarisation [35]. AEA is synthesised in various manners. The best-known pathway is via the enzymes N-acyl phosphatidylethanolamine phospholipase D (NAPE-PLD), which is mostly present in the presynaptic terminal of glutamatergic axon cells [36]. 2-AG is synthesised by diacylglycerol lipase (DAGL). After synthesis, the endocannabinoids travel in a retrograde manner [37] to bind on the CB1 receptors on the presynaptic membrane where, once activated, CB1 receptors inhibit adenylyl cyclase activity, which reduces the cyclic adenosine monophosphate (cAMP) cascade and, in turn, augments the activation of potassium channels, which inhibit calcium influx via voltage-gated calcium channels [38]. Subsequently, neurotransmitter release is inhibited and termination of the action of AEA and 2-AG takes place via enzymatic hydrolysis [39]. Fatty acid amide hydrolase (FAAH) is the catabolic enzyme that hydrolyses AEA, while monoacylglycerol (MAGL) hydrolyses 2-AG [39,40].

So far, only one genetic polymorphism related to the ECS has been found to be reliably involved in fear and anxiety-related processes: the *FAAH* polymorphism C385A, rs324420. Approximately 38% of the European population are A-allele carriers (AA/AC), leading to the FAAH enzyme being more vulnerable to a proteolytic degradation, which, in case of AA-homozygotes, leads to less than half of the FAAH activity as compared to CC-homozygotes and thus higher peripheral AEA levels [41,42,43,44]. In line with these findings, animal studies were able to demonstrate enhanced fear extinction processes when using a FAAH inhibitor [45]. Studies indicate lower anxiety levels in human A-allele carriers [28,43,46]. Recently, Mayo et al. (2018) have found that gene-dependent higher levels of AEA enhanced fear extinction and extinction recall, and that elevated AEA levels in AA-homozygotes have a protective function during human stress-related responses, in a way that A-allele homozygotes showed no significant decreases in AEA levels after a stress task [44]. In another study conducted by our research group, we found differential human brain activation upon an unextinguished relative to an extinguished stimulus, which was greater in AC heterozygotes as compared to CC homozygotes in core neural structures previously related to extinction recall, pointing towards more successful extinction recall in human A-allele carriers with elevated AEA levels [25].

In recent years, many studies have focussed on the effects of the ECS on PTSD, with AEA being the most promising target. Interestingly though, the results have been mixed. Hauer et al. (2013) found higher plasma AEA concentrations in PTSD patients and interpreted the increased endocannabinoid activity to control homeostasis [47]. On the contrary, Neumeister et al. [48] found reduced peripheral AEA levels in PTSD patients, which complemented their findings of upregulated brain-wide CB1 receptor availability. However, their results might have been confounded by a significantly higher smoking frequency in the PTSD group.

In patients with BPD, previous studies showed diverging results. Schaefer et al. (2014) found higher AEA and 2-AG serum concentrations in patients as compared to a control group. The authors suggested that the elevated AEA levels might attenuate depressive symptomatology, while elevated levels of 2-AG were interpreted as analgetic [31]. On the other hand, Wingenfeld et al. (2018) found decreased AEA-concentrations in hair samples, however, they correlated negatively with the occurrence of childhood traumata as well as depression, pointing towards lower trait endocannabinoid concentrations in BPD and an influence of early aversive events [32]. In addition, Kolla et al. (2020) conducted a PET radiotracer study and found elevated *FAAH*-levels in BPD patients in the prefrontal cortex and the amygdala [49].

Given that the ECS is an easily accessible psychopharmacotherapeutic target, and that many patients frequently self-medicate cannabis for its anxiolytic, analgetic and calming effect, it has become a central topic of current research [50,51,52] and compounds targeting the ECS might serve as an adjuvant treatment component in BPD therapy. Psychotherapeutic interventions have been shown to successfully reduce BPD symptom severity [53], however they are limited and require an immense motivation on the side of the patient. Psychopharmacotherapeutic options such as antidepressants, sedatives and antipsychotic drugs are widely used, however research results do not support their effectiveness [54]. Given the lack of effective treatment options, investigating potentially effective transmitter systems could be beneficial in the treatment of BPD.

Here, we investigate endocannabinoid concentrations (AEA and 2-AG) as well as the *FAAH* SNP C385A in a large sample of patients with BPD and healthy controls, to investigate potential imbalances and links to psychopathology.

## 2. Results

Patients and HCs were successfully matched with no group differences regarding age, gender, body mass index and smoking status. As expected, patients scored significantly higher in the questionnaires BDI, BSL and STAI-S compared to HCs. Regarding endocannabinoid levels, we found significantly higher plasma AEA levels and by trend higher plasma 2-AG levels in the patients compared to HCs (see Table 1 and Figure 1). To further investigate these differences, we calculated effects of genotype and correlations between endocannabinoid levels and psychopathology measures. For details on demographic data see Section 4.

Regarding the *FAAH* polymorphism (rs324420), proportions of CC genotype and A-allele carriers did not differ between groups. However, the expected higher anandamide (AEA) levels in A-allele carriers (M = 0.85/SD = 0.35 pmol/mL) as compared to the CC genotype group (M = 0.56/SD = 0.16 pmol/mL) were only observed in HCs (t(30) = 3.03, *p* = 0.002). In patients, mean AEA levels of M = 1.11/SD = 0.58 were observed in A-allele carriers and of M = 0.95/SD = 0.54 pmol/mL in the CC genotype group, which did not differ significantly (t(47) = 0.89, *p* = 0.17). Accordingly, the observed elevated AEA levels in patients relative to HCs irrespective of the *FAAH* polymorphism (rs324420) seem to be more related to elevated levels in patients with the CC genotype, where we found a significant difference to controls with the CC genotype (t(48) = 2.92, *p* = 0.005). AEA concentrations in A-allele carrier patients, did not differ significantly from those in controls (t(29) = 1.49, *p* = 0.15). Correction for one outlier in the BPD-CC group did not alter the significances of these results. See also Figure 1 and Table 1.

Exploratory ANOVAs with factors ‘diagnosis’ (patient/control) and ‘*FAAH*_rs32422 genotype’ (CC/CA-AA) on STAI-S, BSL and BDI-II confirmed significant main effects of diagnosis for STAI-S, BSL and BDI-II (*p* < 0.001). No significant main effects of genotype were found for STAI-S and BSL, but for the BDI-II (F = 5.06, *p* = 0.028). Higher values of depression were found in subjects with the CC genotype as compared to A-allele carriers, particularly in the patient group (see Figure 2). Interactions of factors ‘diagnosis’ and ‘genotype’ were not significant. An ANOVA with the same factors (diagnosis, genotype) excluded effects of BMI, as no main effect of diagnosis nor genotype was found for the BMI (F = 1.18, *p* = 0.28).

No correlations were found between psychometric measures (BSL, STAI-S and BDI-II) and AEA or 2-AG-levels.

## 3. Discussion

Our study reveals significant variations in endocannabinoid concentrations in patients with borderline personality disorder (BPD) and healthy controls (HC). Specifically, we observed higher state anandamide (AEA) concentrations among BPD patients as compared to the HC group. A similar trend was observed for 2-AG, although statistical significance was not reached. These findings align with previous research, including Schaefer et al. (2014) [31], who reported elevated AEA and 2-AG concentrations in a BPD sample and other studies suggesting the involvement of the endocannabinoid system (ECS) in anxiety regulation [35,43,55,56].

Upon closer examination of the genetic variation regarding the metabolizing enzyme *FAAH* (SNP rs324420), the commonly [35,43] observed increased AEA levels in A-allele carriers as compared to the CC genotype were evident in the control, but not in the patient group. This effect of a reduced difference between CC and CA/AA genotype AEA levels in BPD might relate to relatively increased AEA in patients with the CC genotype. In line with this interpretation, patients with the CC genotype also exhibited a substantial elevation in state AEA levels compared to the CC carriers in the HC group. No significant differences between HC and BPD were found when comparing A-allele carriers with CA/AA genotypes. This observation may further contribute to the understanding of ECS mechanisms. On demand ECS reactivity regarding anxiety and/or depression is well known, [34,55,57] and elevated AEA concentrations in general have been linked to antidepressant and anxiolytic effects [35,57,58]. The finding could be interpreted in a way that persistent or temporary compensatory ECS activity is particularly evident in the CC genotype as compared to A-allele carriers to balance physiologically lower AEA levels. The observed higher depression ratings in the CC genotype might corroborate a potential demand for such a compensatory, but potentially yet insufficient elevation. While Schaefer et al. (2014) found generally higher AEA levels in patients compared to HCs [31], the elevation in our study, particularly in CC-allele carriers with higher depression values may support the hypothesis of dynamic changes in peripheral AEA according to individual demands and baseline levels [24].

Previous findings regarding ECS alterations in MDD and PTSD are mixed. Behnke et al. (2023) found higher plasma AEA levels in women with MDD, which could align with the hypothesis of an elevation to compensate for depressive symptoms [59], while Walther et al. (2023) found a negative correlation between MDD severity and AEA levels in hair [60]. However, the relation between plasma and hair endocannabinoid levels is unclear. The same accounts for PTSD, where increased peripheral AEA concentrations have been found by Hauer et al. (2013) [47], while Neumeister et al. (2013) found lower levels [48]. However, none of these studies examined any genetic variants.

The findings of Wingenfeld et al., who identified a trendwise negative correlation between self-reported depression severity in BPD patients and AEA concentrations, are in line with our finding of elevated depression scores in the CC genotype, indicating that lower AEA levels are linked to more severe symptoms of depression [32]. However, they found generally lower AEA concentrations in BPD patients when analysing hair samples, suggesting that different mechanisms may be at play concerning short-term (as measured in plasma) and long-term (as measured in hair samples) endocannabinoid availability. Furthermore, in a female-specific sample, Hill et al. (2009) reported lower serum endocannabinoid levels in depressed patients compared to a control group [58]. Although our data replicate the findings of Schäfer et al. (2014) regarding BPD and plasma endocannabinoid levels [31], it is crucial to stress the need for replication studies to confirm the connection between BPD, depression and the ECS.

It is well-established that the ECS plays a vital role in brain development during childhood and adolescence [61], a period from which vulnerability for BPD originates. This is related to socioenvironmental factors and early trauma or adverse childhood experiences shaping individuals’ personalities and interactive behaviours. While traumatic experiences in childhood and adolescence may lead to a chronically elevated allostatic load, resulting in persistent elevation of endocannabinoids, our data indicate that current depression and interindividual differences might play a greater role in ECS alterations.

Regarding depression, pharmaceutical investigations show compelling evidence for an involvement of the ECS. When first introduced, the CB1 receptor antagonist Ribonamant was linked to an increased risk for the development of depression by inducing anhedonia [62]. In line, enhanced CB1 receptor signalling has been shown to have antidepressant effects in rodents, similar to conventional antidepressants [56,63]. These findings could be relevant for the future development of new pharmacotherapeutic drugs addressing both anxiety and depressive symptoms, which are both very common in BPD. The observed increased AEA concentrations might suggest a putative role for modulators of the ECS in the treatment of BPD by restoring disrupted homeostasis and alleviate the effects of (chronic) stress and depressive illness. However, it remains unclear whether elevated AEA levels are indeed a consequence of BPD symptoms as increased AEA levels could also be involved in the development of the disorder. Here, future studies should emphasise mapping ECS alterations during childhood and adolescence, to further elucidate the development of pathological mechanisms, which may offer insights into appropriate treatments. When considering the ECS for the development of new adjuvant pharmacotherapeutic measures, elevated endocannabinoid levels might initially serve a protective role in restoring homeostasis, however, it is crucial to consider limiting endocannabinoid elevation, as for example during trauma exposure, to ensure therapeutic efficacy while minimising potential long-term risks. These risks have not been fully investigated, but chronically elevated endocannabinoids have been associated with increased levels of pro-inflammatory markers [64] which might for example partly explain the heightened mortality observed in PTSD patients [65], consistent with increased AEA and 2-AG concentrations in patients with coronary artery disease [64]. However, it is essential to acknowledge that these interpretations remain speculative.

Our findings should be interpreted considering certain limitations, including uneven sample sizes and gender distribution. As one of the few studies investigating endocannabinoids in BPD, we did not control for individual menstrual cycle variations, which could have influenced our results. Additionally, most of the patients in our study underwent psychopharmacological treatment with antidepressants at the time of the study, which may have impacted our findings although current research does not suggest influences of antidepressants on endocannabinoid levels. However, future research should replicate this study in medication-free patients with BPD to investigate alterations in patients with and without psychopharmacological treatment.

Despite these limitations, our findings show that BPD is associated with elevated AEA concentrations, similar to Schaefer et al. (2014) [31], pointing towards altered endocannabinoids in BPD and highlighting the role of the *FAAH* SNP rs324420. The data could be interpreted in the way of a compensatory elevation, particularly in the CC genotype. However, it is essential to recognize that multiple parameters such as traumatic experiences in early life, depression or other comorbid disorders and factors such as NSSI, suicidal ideation, addiction or drug and alcohol abuse may affect endocannabinoid disruptions differently. Future studies should investigate these effects in a larger BPD sample, both before and after treatment. These efforts will be crucial for understanding the ECS’s role in BPD and its potential for therapeutic interventions.

## 4. Materials and Methods

The study was conducted according to the guidelines of the Declaration of Helsinki and approved by the Ethics Committee of Ulm University, Germany (#221/21). All participants signed an informed consent before being included.

### 4.1. Participants

All patients who took part in the 8-week DBT inpatient program for BPD at the Department of Psychiatry and Psychotherapy III of Ulm University Hospital between July 2021 and July 2022 were offered the choice of participating in the study. Of 73 patients admitted during this period to the program, 54 patients with a BPD diagnosis participated in the study. Diagnoses of BPD according to ICD-10 were assessed with the “International Personality Disorder Examination” (IPDE) [66] by a trained psychiatrist. Most common comorbid diagnoses according to ICD-10 were Major Depression (MD) in 38 patients, PTSD (21 patients) and Mental and Behavioural Disorders due to alcohol (at the time of assessment: 4 patients; in the past: 13 patients) or cannabinoids (at the time of assessment: 1 patient; in the past: 13 patients). Urine screenings for cannabis use obtained upon admission to the program confirmed abstinence at the time of the study. A total of 3 patients and none of the healthy controls (HC) had used cannabis within the 4 weeks before the investigation. HCs were recruited in Ulm by means of flyers and word of mouth advertising. HCs were matched to patients regarding gender, age, body mass index (BMI) and nicotine use. Current or lifetime Axis I disorder was excluded by screening all control subjects with a Structured Clinical Interview for Diagnosis—Axis I (SCID-I). We present data from 32 HCs and the 49 of the total sample of 54 patients from which endocannabinoid data could be obtained upon admission (see section: Blood sampling). A detailed characterisation of the sample is shown in Table 2.

### 4.2. Blood Sampling

Blood was taken under fasting conditions in the morning in all participants. Blood samples from patients were obtained at up to three time points at which blood was taken for clinical routine, i.e., upon admission, in the middle of the program after about 4 to 5 weeks and upon discharge after about 7 to 10 weeks. In healthy controls, only one blood sample was obtained. In this paper, we present endocannabinoid data from blood samples obtained upon admission. We investigated serum levels of AEA and 2-AG by mass spectrometry analysis blood-to-plasma processing. Extraction and analysis of plasma endocannabinoids were carried out following previously described protocols [67].

### 4.3. Questionnaires

BPD symptom severity upon admission and upon discharge in the patients was measured with the short version of the Borderline Symptom List (BSL-23) [68]. The self-rating instrument with good psychometric properties consists of 23 statements, with which patients rate their agreement using a 5-point Likert scale ranging from “not at all” to “very much”. Here, we analyse the mean score on the BSL-23 as the patient-reported outcome. Depressive symptoms before and after treatment were measured with the German version of the Beck Depression Inventory, second edition (BDI-II, REF; German version [69]). The State Trait Anxiety Inventory for Adults (STAI, German Version [70] was used to assess patients’ state anxiety (STAI-S scores) after each blood draw, i.e., up to three times in each patient. In healthy control subjects, BSL-23, BDI-II and STAI-S scores were obtained directly before or after the blood draw.

### 4.4. Statistics

Calculations were performed using Microsoft Excel 2016 MSO 64-Bit and Statistica Version 13, TIBCO Software Inc. for Windows. Comparisons between groups (patients/controls) were calculated using X^2^-tests or independent sample *t*-tests (two-tailed) where appropriate. Elevated anandamide levels in A-allele carriers were confirmed using one-tailed *t*-tests. ANOVAS with factors ‘diagnosis’ and ‘genotype’ were used to exploratorily investigate the effects of genotype on questionnaire data and body mass index.

### 4.5. FAAH-Genotyping

DNA was extracted applying standard protocols of a commercial extraction kit (MagNA Pure 96 DNA and Viral NA Small Volume Kit) and the MagNA Pure 96 (Roche Diagnostics, Mannheim, Germany). Genotyping at *FAAH* C385A was performed via real-time quantitative polymerase chain reaction using melting curve detection analysis with Cobas z480 LightCyclers (Roche Diagnostics, Mannheim, Germany). The primers were obtained from TIB-Molbiol, Berlin, Germany.

### 4.6. Mass Spectrometry of Anandamide

Plasma levels of the endocannabinoid N-arachidonoylethanolamide (AEA) were assessed using mass spectrometry. For each participant, 7.5 mL EDTA blood samples were directly centrifuged (10 min, 2000× *g*, 4 °C) and 100 µL of blood plasma was directly pipetted and stored at −30°C for one week and then stored at −80 °C for approximately 4 weeks.

Endocannabinoids were extracted from 100 µL human plasma by liquid-liquid-extraction (LLE). Plasma samples were allowed to thaw on ice for 90 min. Then, 250 µL of ethylacetate/n-hexane (9:1, *v*/*v*) and a 50 µL mixture of internal standards in acetonitrile, e.g., AEA-d4, 2-AG-d5 and AA-d8 were then added to the plasma samples. The concentration of internal standards in the spiking mixture was set to render a target concentration of the deuterated standards in the final sample extracts of 0.25 ng/mL for AEA-d4, 2.5 ng/mL for 2-AG-d5, 100 ng/mL for AA-d8, respectively [71]. Samples were vortexed for 30 s and then centrifuged for 5 min at 4 °C and at 5080× *g* with a swing rotor. Following centrifugation, samples were kept at −20 °C for 10 min for phase separation. The upper organic phase was recovered and pipetted in 96 deep-well plates and allowed to evaporate at 37 °C under a gentle stream of nitrogen. The extracts were then reconstituted in 50 µL acetonitrile/water (1:1, *v*/*v*). The solubilized extracts were then transferred in 96 microtiter plates for LC/MRM injection. LC/MRM conditions for analysis and quantification of eCBs were as described in [71,72]. Throughout the entire sample extraction procedure as well as in the LC/MRM autosampler samples were maintained at 4 °C, and additionally all solvents and vials were pre-cooled at 4 °C to prevent and/or standardize ex-vivo changes of eCBs levels across all samples. eCBs values were normalized to plasma volume.

## Figures and Tables

**Figure 1 ijms-24-17452-f001:**
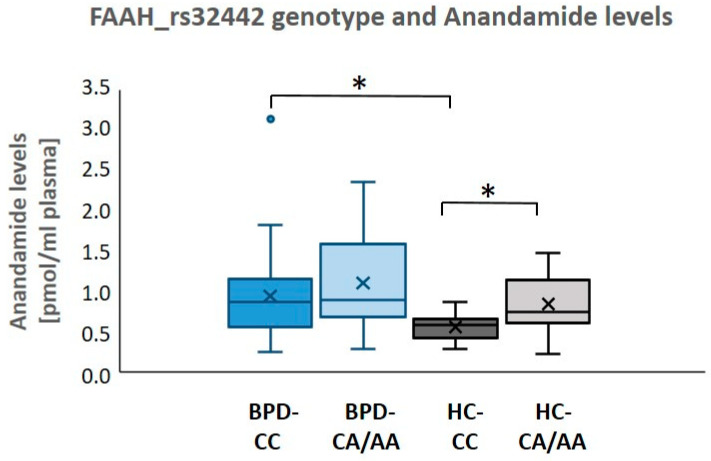
Box–whisker plots show mean values and 25, 50, and 75th percentile of empirical distribution; whiskers extend to smallest and largest value excluding outliers. In healthy controls (HC), significantly higher (*: *p* < 0.005) anandamide (AEA) plasma levels in the CA/AA genotype as compared to CC genotype are demonstrated. In patients with borderline personality disorder (BPD), no difference between CC and CA/AA genotypes was found. Comparing patients with CC genotype (BPD-CC) with HC with CC genotype (HC-CC), we found significantly higher (*: *p* < 0.05) anandamide (AEA) plasma levels while no differences between HC and BPD were found when comparing A-allele carriers with CA/AA genotypes.

**Figure 2 ijms-24-17452-f002:**
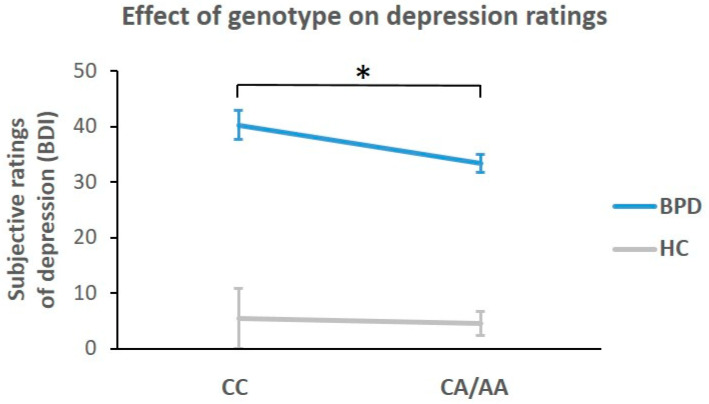
Subjective ratings of depression (Beck Depression Inventory, mean and 95% confidence intervals) and significant (*: *p* < 0.05) effect of genotype with higher ratings in the CC genotype compared to A-allele carriers (*FAAH*_rs32442 genotype). No interaction effect was evident between patients with borderline personality disorder (BPD) and healthy controls (HC).

**Table 1 ijms-24-17452-t001:** Endocannabinoid concentrations and *FAAH* rs324420 distribution.

	Patients (N = 49)	Healthy Controls (N = 32)	Group Differences
Gender	42 females (85.7%)	27 females (84.4%)	X^2^ = 0.03, *p* = 0.87 ^b^
*FAAH*_rs324420: CC/CA/AA	N = 32/14/3	N = 18/13/1	CC vs. CA/AA: X^2^ = 0.67, *p* = 0.41 ^b^
AEA levels, pmol/mL plasma (M ± SD)	1.00 ± 0.55	0.68 ± 0.29	*t*(79) = 2.96, *p* = 0.001 ^a^
2-AG levels, pmol/mL plasma (M ± SD)	1.48 ± 1.69	0.92 ± 0.56	*t*(79) = 1.80, *p =* 0.08 ^a^

Notes: ^a^ independent sample *t*-test; ^b^ chi-square-test; 2-AG = 2-arachidonoylglycerol; AEA = anandamide; M = mean; SD = standard deviation.

**Table 2 ijms-24-17452-t002:** Demographic variables, questionnaire results.

	Patients (N = 49)	Healthy Controls (N = 32)	Group Differences
Gender	42 females (85.7%)	27 females (84.4%)	X^2^ = 0.03, *p* = 0.87 ^b^
Age (M ± SD)	27.02 ± 10.04 [18–61 y]	25.06 ± 3.70 [18–35 y]	*t*(79) = 1.05, *p* = 0.29 ^a^
Body Mass Index: kg/m^2^ (M ± SD)	28.76 ± 8.19	26.93 ± 6.89	*t*(78) = 1.03, *p* = 0.30 ^a^
Smoker (nicotine)	N = 24 (49.0%)	N = 15 (46.9%)	X^2^ = 0.03, *p* = 0.85 ^b^
BDI-II (M ± SD)	37.11 ± 8.53	5.03 ± 3.38	*t*(72) = 20.09, *p* < 0.001 ^a^
BSL sum (M ± SD)	50.18 ± 16.02	2.91 ± 2.37	*t*(75) = 16.54, *p* < 0.001 ^a^
Self-injury (number in the past year, M ± SD)	60.31 ± 95.23		
Number of suicide attempts (lifetime, M ± SD)	3.11 ± 4.05		
Days per week with suicidal thoughts in the past 6 months	4.31 ± 2.36		
Current psychotropic medication	Antidepressants N = 41 Antipsychotics N = 11		
STAI-S sum score at blood draw (M ± SD)	54.88 ± 9.34	30.47 ± 7.18	*t*(78) = 12.51, *p* < 0.001 ^a^

Notes: ^a^ independent sample *t*-test; ^b^ chi-square test; M = mean; SD = standard deviation; STAI-S, Body Mass Index: missing data in 1 patient; BDI: missing data in 7 patients; BSL: missing data in 4 patients.

## Data Availability

Data are contained within the article.

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
