# Peer review of "Plasma Endocannabinoid Levels in Patients with Borderline Personality Disorder and Healthy Controls"

_ijms, 2023, doi:10.3390/ijms242417452_

Round 1

Reviewer 1 Report

Comments and Suggestions for Authors

In the present study, Spohrs and colleagues aimed to evaluate changes in the levels of the main endocannabinoid compounds in female patients with borderline personality disorder (BPD) versus controls, as biomarkers of this psychiatric condition. Is a very interesting and important study in the field, considering the available information. However, some important points need to be addressed before considering the publication of this manuscript in the International Journal of Molecular Sciences journal.

Major points:

1.    Lines 60-61 à the endocannabinoid system also includes the enzymes responsible of the synthesis and degradation of the endocannabinoid compounds. Please, clarify this point in the manuscript.

2.    There is not supportive information about the involvement and the selection of the FAAH SNP C385A, even its strong involvement in different psychiatric conditions. Please, add the corresponding information in the Introduction of the manuscript.

3.    It could be interesting to have also a figure with 2-AG levels in the manuscript, to emphasize the absence of differences between BPD and HC, under these experimental conditions and this number of subjects.

4.    Considering that BPD patients have been treated at the moment of the evaluation, how can the authors ensure that the observed changes are not due to antidepressant or antipsychotic medication?

5.    I consider that the affirmation about the homeostatic effects of the AEA increased levels should be considered with caution, and this is a statement that is not based on the results observed in the study.

6.    It could be interesting to carry out this quantification in medication-free BPD patients, to compare the alterations in the endocannabinoid system in those patients with and without treatment and further explore the involvement of this systems in the psychopathology of the BPD.

7.    Under my point of view, more emphasis is necessary on the results of this study and not on assumptions and results of other studies in the Discussion section.

Minor points:

1.    Please, revise the sentence in lines 131-133, apparently some information is missing which difficult the understanding.

2.    Please, revise all the references, some of them are not adequately inserted in the manuscript (lines 169-170).

Comments on the Quality of English Language

Some English revisions are needed. 

Author Response

Reviewer 1

In the present study, Spohrs and colleagues aimed to evaluate changes in the levels of the main endocannabinoid compounds in female patients with borderline personality disorder (BPD) versus controls, as biomarkers of this psychiatric condition. Is a very interesting and important study in the field, considering the available information. However, some important points need to be addressed before considering the publication of this manuscript in the International Journal of Molecular Sciences journal.

Dear reviewer 1,

Thank you very much for the considerate highlighting of the strengths of our work and also the highly valuable remarks that have been very helpful to further improve the manuscript. We have edited the sections accordingly.

In the following, we would like to address the comments separately:

Major points:

Comment 1.    Lines 60-61 à the endocannabinoid system also includes the enzymes responsible of the synthesis and degradation of the endocannabinoid compounds. Please, clarify this point in the manuscript.

Comment 2.    There is not supportive information about the involvement and the selection of the FAAH SNP C385A, even its strong involvement in different psychiatric conditions. Please, add the corresponding information in the Introduction of the manuscript.

Response 1.&2.) Thank you for these comments. We have added the relevant information on synthesis, degradation and the SNP.

Comment 3.    It could be interesting to have also a figure with 2-AG levels in the manuscript, to emphasize the absence of differences between BPD and HC, under these experimental conditions and this number of subjects.

Response 3.) Since AEA and 2-AG levels are displayed in Table 2, we opted against duplicating the presentation of 2-AG data in figure form. Since no SNPs regarding 2-AG were assessed, a figure for 2-AG paralleling figure 1 regarding AEA and FAAH-genotype cannot be designed, unfortunately.

Comment 4.    Considering that BPD patients have been treated at the moment of the evaluation, how can the authors ensure that the observed changes are not due to antidepressant or antipsychotic medication?

Comment 6.    It could be interesting to carry out this quantification in medication-free BPD patients, to compare the alterations in the endocannabinoid system in those patients with and without treatment and further explore the involvement of this systems in the psychopathology of the BPD.

Response 4.&6.) Thank you very much for these interesting remarks. Given the current literature in the field, we find no evidence that suggests such an effect of medication on endocannabinoid concentrations. However, we fully agree with you that such effects cannot be excluded and added this limitation and impulse for future research to the discussion section.

Comment 5.    I consider that the affirmation about the homeostatic effects of the AEA increased levels should be considered with caution, and this is a statement that is not based on the results observed in the study.

Response 5.) Thank you for making us aware of this. Interpretations in Abstract and Discussion regarding the potential function of the increased AEA levels were toned down and are worded with more caution, as suggested.

Comment 7.    Under my point of view, more emphasis is necessary on the results of this study and not on assumptions and results of other studies in the Discussion section.

Response 7.) Thank you for pointing this out. We have thoroughly revised the discussion accordingly.

Minor points:

Comment 1.    Please, revise the sentence in lines 131-133, apparently some information is missing which difficult the understanding.

Response 1.) The sentence has been revised. Thank you for highlighting it.

Comment 2.    Please, revise all the references, some of them are not adequately inserted in the manuscript (lines 169-170).

Response 2.) All the references have been checked. Thank you for pointing it out.

English revisions: The manuscript has been prove read by a native speaker.

Reviewer 2 Report

Comments and Suggestions for Authors

The authors describe a study in which they analyzed plasma concentrations of anandamide (AEA) and 2-arachi-15 donoylglycerol (2-AG), in female BPD patients and 32 matched healthy controls. They also analyzed the impact of the FAAH polymorphism rs324420 and correlations with psychopathology. Compared to controls, the patients showed higher AEA levels and a trend towards higher 2-AG levels; Carriers of the CC allele pattern (who produce less AEA) showed higher AEA levels (In the control group only) and higher depression levels (only in the patient group), compared to those with other allele patterns. The endocannabinoid results are interpreted as representing compensation (for psychopathology?) of the physiology towards homeostasis.

Comments

Abstract

The results regarding the allele differences should be described more clearly. This: " The effect was particularly related to 19 a substantial elevation among patients with the CC-genotype. An effect of genotype was found for 20 current depression values as assessed with the Beck’s depression inventory (BDI-II) with higher 21 values in patients with the CC genotype as compared to A allele carriers" does not describe the findings precisely.

Introduction

This section presents the topic clearly, describing the conflicting findings in this field and the aims of the study.

Methods

Where and how was the healthy control sample obtained?

Results

I suggest that the endocannabinoid results part of Table 1 (last 3 items) should be taken from there and presented separately in a different table in the Results section.

Discussion

" it became evident that higher state AEA concentrations were particularly pronounced in individuals with the CC genotype within the patient group, while the commonly [36], [38] observed increased AEA levels in A-allele carriers as compared to the CC genotype were not evident. "

-          This summary of results does not seem to fit with the actual results, or maybe I missed something.

-          A) "it became evident that higher state AEA concentrations were particularly pronounced in individuals with the CC genotype within the patient group" – according to Figure 1, this was only significant in the control group, and in the opposite direction – AEA levels were LOWER in the CC phenotype.

 -          B) "while the commonly [36], [38] observed increased AEA levels in A-allele carriers as compared to the CC genotype were not evident. – In the control group it seems to be evident (Figure 1).

Beyond this, the Discussion section is overall fine, clearly discussing the findings within the existing literature and clearly aware of the study's limitations.

Author Response

Reviewer 2

The authors describe a study in which they analyzed plasma concentrations of anandamide (AEA) and 2-arachi-donoylglycerol (2-AG), in female BPD patients and 32 matched healthy controls. They also analyzed the impact of the FAAH polymorphism rs324420 and correlations with psychopathology. Compared to controls, the patients showed higher AEA levels and a trend towards higher 2-AG levels; Carriers of the CC allele pattern (who produce less AEA) showed higher AEA levels (In the control group only) and higher depression levels (only in the patient group), compared to those with other allele patterns. The endocannabinoid results are interpreted as representing compensation (for psychopathology?) of the physiology towards homeostasis.

Dear Reviewer 2,

thank you very much for the time spent on reviewing our article and the positive feedback.

We have edited our article according to your feedback and believe that it has improved the quality of the manuscript.

Following, we would like to address your comments point by point:

Abstract:

Comment: The results regarding the allele differences should be described more clearly. This: "The effect was particularly related to a substantial elevation among patients with the CC-genotype. An effect of genotype was found for current depression values as assessed with the Beck’s depression inventory (BDI-II) with higher values in patients with the CC genotype as compared to A allele carriers" does not describe the findings precisely.

Response: We have rewritten the Results section of the abstract and hope that it describes the results more precisely, now.

Methods:

Comment: Where and how was the healthy control sample obtained?

Response: Thank you very much for pointing this out! We have added information on the study sample.

Results:

Comment: I suggest that the endocannabinoid results part of Table 1 (last 3 items) should be taken from there and presented separately in a different table in the Results section.

Response: Excellent suggestion; we have added another table merely for the endocannabinoid data.

Discussion:

Comment: " it became evident that higher state AEA concentrations were particularly pronounced in individuals with the CC genotype within the patient group, while the commonly [36], [38] observed increased AEA levels in A-allele carriers as compared to the CC genotype were not evident. "

-          This summary of results does not seem to fit with the actual results, or maybe I missed something.

-          A) "it became evident that higher state AEA concentrations were particularly pronounced in individuals with the CC genotype within the patient group" – according to Figure 1, this was only significant in the control group, and in the opposite direction – AEA levels were LOWER in the CC phenotype.

 -          B) "while the commonly [36], [38] observed increased AEA levels in A-allele carriers as compared to the CC genotype were not evident. – In the control group it seems to be evident (Figure 1).

Response: Thank you for the comment. Indeed, our wording was not clear enough. As now stated clearly in text and figure legend, we find significantly higher anandamide (AEA) plasma levels in the CA/AA genotype as compared to CC genotype in healthy controls. In patients with borderline personality disorder (BPD), no difference between CC and CA/AA genotypes was found. Comparing patients with CC genotype (BPD-CC) with control persons with CC genotype (HC-CC), we found significantly higher anandamide (AEA) plasma levels, while no differences between HC and BPD were found when comparing A-allele carriers with CA/AA genotypes. We apologize for misleading wording and hope that the changes made render the text more understandable.

Round 2

Reviewer 1 Report

Comments and Suggestions for Authors

The authors have responded adequately to my previous questions and under my consideration, the manuscript is ready for publication.